# Directed gas phase formation of silicon dioxide and implications for the formation of interstellar silicates

Tao Yang [1,2], Aaron M. Thomas[1], Beni B. Dangi[1,3], Ralf I. Kaiser [1], Alexander M. Mebel [4] & Tom J. Millar [5]

Interstellar silicates play a key role in star formation and in the origin of solar systems, but their synthetic routes have remained largely elusive so far. Here we demonstrate in a combined crossed molecular beam and computational study that silicon dioxide ($SiO_2$) along with silicon monoxide (SiO) can be synthesized via the reaction of the silylidyne radical (SiH) with molecular oxygen ($O_2$) under single collision conditions. This mechanism may provide a low-temperature path—in addition to high-temperature routes to silicon oxides in circumstellar envelopes—possibly enabling the formation and growth of silicates in the interstellar medium necessary to offset the fast silicate destruction.

[1] Department of Chemistry, University of Hawai'i at Mānoa, Honolulu, HI 96822, USA. [2] State Key Laboratory of Precision Spectroscopy, East China Normal University, Shanghai, 200062, China. [3] Department of Chemistry, Florida Agricultural and Mechanical University, Tallahassee, FL 32307, USA. [4] Department of Chemistry and Biochemistry, Florida International University, Miami, FL 33199, USA. [5] Astrophysics Research Centre, School of Mathematics and Physics, Queen's University Belfast, Belfast, BT7 1NN, UK. Correspondence and requests for materials should be addressed to R.I.K. (email: ralfk@hawaii.edu) or to A.M.M. (email: mebela@fiu.edu) or to T.J.M. (email: tom.millar@qub.ac.uk)

The origin of interstellar silicate grains—nanoparticles consisting primarily of olivine-type $((Mg,Fe)_2SiO_4)$ refractory minerals—has remained a controversial topic for more than half a century, since interstellar silicates are faster destroyed by sputtering than formed during the late stages of stellar evolution through nucleation in circumstellar envelopes of oxygen-rich Asymptotic Giant Branch (AGB) and Red Supergiant (RSG) stars[1–5]. These nanoparticles have been associated with the prebiotic evolution of the interstellar medium (ISM) through the synthesis of molecular building blocks of life such as amino acids and sugars on their ice-coated surfaces by ionizing radiation[6]. Interstellar silicates also play a critical role in star formation and in the origin of solar systems contributing to the radiation balance and acting as a molecular feedstock, both through the formation of complex organics through the release of icy mantles that cover them and through disruption of grains in interstellar shocks[3,7]. In molecular clouds, they absorb light and hence shield complex organic molecules (COMs)—organics containing carbon, hydrogen, nitrogen, and oxygen like glycolaldehyde $(HCOCH_2OH)$ and formamide $(HCONH_2)$—from the destructive interstellar ultraviolet radiation field[8]. Therefore, the elucidation of the origin of interstellar silicates is of vital importance to the astrochemistry, astrobiology, and astrophysics communities to eventually understand the fundamental processes that create a visible galaxy including our own.

A crucial point of concern is that the mass of dust ejected during the late stages of stellar evolution is produced at a rate that is significantly slower than the dust destruction time in the ISM, implying that grains also form in the lower density environment of the ISM[5,9–14]. Current astrochemical models of circumstellar envelopes propose that dust formation in AGB stars is driven eventually by clustering and reactions of silicon oxides along with magnesium-type and iron-type oxides[2,15–20]. There does appear to be, however, a severe discrepancy between the formation rates of silicate grains in circumstellar envelopes of $3 \times 10^9$ years and their destruction via sputtering once dispersed into the ISM that limits their lifetime to only a few $10^8$ years[5,21–23]. This discrepancy[24] may eventually be resolved through a better understanding of the processes of dust destruction[5,23], but it remains possible that significant formation of dust needs to occur in the interstellar as opposed to the circumstellar medium[12–15]. Indeed, silicate grains may grow in the ISM by accreting and incorporating silicon oxide molecules[25,26].

Here we show that the silicon dioxide molecule $(SiO_2)$ along with silicon monoxide (SiO) can be efficiently formed via a low-temperature gas phase chemistry even at 10 K. We report the results of a combined crossed molecular beam study and of electronic structure calculations on the reaction of the D1-silylidyne radical (SiD; $X^2\Pi$) with molecular oxygen $(O_2, X^3\Sigma_g^-)$ leading to the formation of $SiO_2$ and SiO through a barrierless reaction[27]. This system represents a proxy for the reaction of the silylidyne radical (SiH) generated via photolysis of silane $(SiH_4)$[28–30] with $O_2$ to synthesize silicon oxides via a single collision event. In the ISM, the reaction of SiH with $O_2$ may represent a potential pathway to $SiO_2$ and silicon oxide formation in those molecular clouds, where gas phase chemistry follows ice mantle sublimation; these silicon oxides might drive an exothermic chemistry that possibly produces larger silicon oxides[25,31] leading ultimately to silicates at low temperatures.

## Results

**Crossed molecular beam studies in the laboratory frame.** The reactive scattering experiments were performed using a crossed molecular beam apparatus (Methods). We monitored the scattering signal at mass-to-charge ratio ($m/z$) 62 $(^{28}SiDO_2^+)$, 60

$(^{28}SiO_2^+)$, and 44 $(^{28}SiO^+)$. No signal was observed at $m/z$ 62, suggesting that under single collision conditions the lifetime of the $^{28}SiDO_2$ adduct is shorter than its flight time to the electron-impact ionizer. Reactive scattering signal was detected at $m/z$ 60 $(^{28}SiO_2^+)$ (Fig. 1). The scattering signal is relatively weak, and at each angle up to $6 \times 10^6$ time-of-flight (TOF) spectra (60 h collection time) had to be averaged to obtain a reasonable signal-to-noise ratio. Signal detection at $m/z$ 60 alone provides conclusive evidence on the formation of a molecule with the formula $^{28}SiO_2$ via a single collision event of two neutral reactants. Taking into account the data accumulation time, the signal-to-noise ratio obtained at $m/z$ 60 and the abundances of naturally occurring silicon isotopes of $^{30}Si(3.10 \%)$, $^{29}Si$ (4.67 %), and $^{28}Si(92.23 \%)$, we would not expect—as confirmed experimentally—any reactive scattering signal at $m/z$ 62 $(^{30}SiO_2^+)$. Finally, the background counts at $m/z$ 44 originating from singly ionized carbon dioxide $(CO_2)$ in the detector precludes an identification of any reactive scattering signal at $m/z$ 44 $(^{28}SiO^+)$. To summarize, the laboratory data indicate that a molecule with the formula $^{28}SiO_2$ (hereafter: $SiO_2$) along with atomic deuterium is formed under single collision conditions in the reaction of the SiD radical with $O_2$.

**Crossed molecular beam studies in the center-of-mass frame.** We transformed the experimental data from the laboratory to the center-of-mass (CM) reference frame[32] to gain information on the underlying reaction dynamics, which yields the CM translational energy flux distribution $P(E_T)$ and the CM angular flux distribution $T(\theta)$ as depicted in Fig. 2. Best fits of the laboratory data are achieved with a single-channel fit forming products with a mass combination of 60 amu $(SiO_2)$ and 2 amu (D) (Figs. 1 and 2). A detailed inspection of the CM functions affords crucial information on the pertinent reaction channel(s) and dynamics. First, $P(E_T)$ assists in the identification of the product isomer(s). For the reaction products formed without internal excitation, the high energy cutoff of $493 \pm 57$ kJ mol$^{-1}$ in $P(E_T)$ denotes the sum of the absolute value of the reaction exothermicity plus the collision energy $E_c$ ($33.2 \pm 2.0$ kJ mol$^{-1}$). A subtraction of the collision energy reveals that the reaction is highly exothermic with the energy of $-460 \pm 59$ kJ mol$^{-1}$. This finding agrees nicely with our computed value of $-441 \pm 5$ kJ mol$^{-1}$ (Fig. 3) and the energetics obtained from NIST Webbook ($-464$ kJ mol$^{-1}$)[33] to form the linear $SiO_2$ molecule ($l$-$SiO_2$) along with atomic hydrogen (H). This shows for the very first time that a $SiO_2$ molecule is observed in the gas phase as a result of a reaction between two neutral species under controlled experimental conditions. Previously, Ahmed et al. generated $SiO_2$ via laser ablation of silicon (Si) and two successive oxygen abstractions from the seeding and reactant gas—$CO_2$;[34] Wang et al. formed gas phase $SiO_2$ through $SiO_2^-$ electron photodetachment[35]. A complex forming reaction mechanism is evident from $T(\theta)$ which depicts the flux over the complete angular range[36]. This distribution reveals further a forward-backward symmetry proposing that the lifetime of the decomposing intermediate is longer than its rotational period[37]. Alternatively, a 'symmetric' reaction intermediate can account for these findings by emitting a deuterium atom with equal probabilities into $\theta$ and $\pi-\theta$[38].

**Electronic structure calculations and reaction mechanism.** We combined the experimental findings with electronic structure calculations on the reaction of SiH with $O_2$ to elucidate the underlying dynamics (Fig. 3). These calculations were performed at a level of theory high enough to predict relative energies of the transition states and local minima as well as reaction energies within 5 kJ mol$^{-1}$ (Methods). Based on the difference in zero

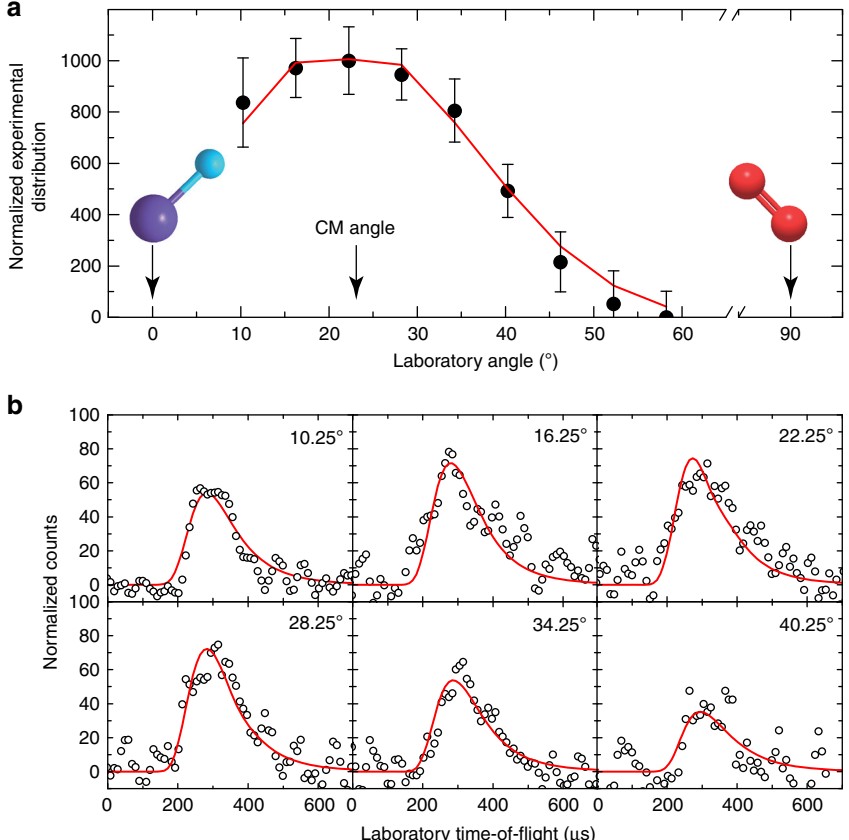

**Fig. 1** Laboratory angular distribution and the associated time-of-flight spectra. Laboratory angular distribution at mass-to-charge ratio of 60 ($SiO_2^+$) recorded in the reaction of the D1-silylidyne radical with molecular oxygen (**a**), and the time-of-flight spectra recorded at distinct laboratory angles overlaid with the best fits (**b**). The solid circles with their error bars indicate the normalized experimental distribution with ±1σ uncertainty (s.d. of the integrals of the time-of-flight spectra for the respective angle), and the open circles indicate the experimental data points of the time-of-flight spectra. The red lines represent the best fits obtained from the optimized center-of-mass (CM) functions, as depicted in Fig. 2

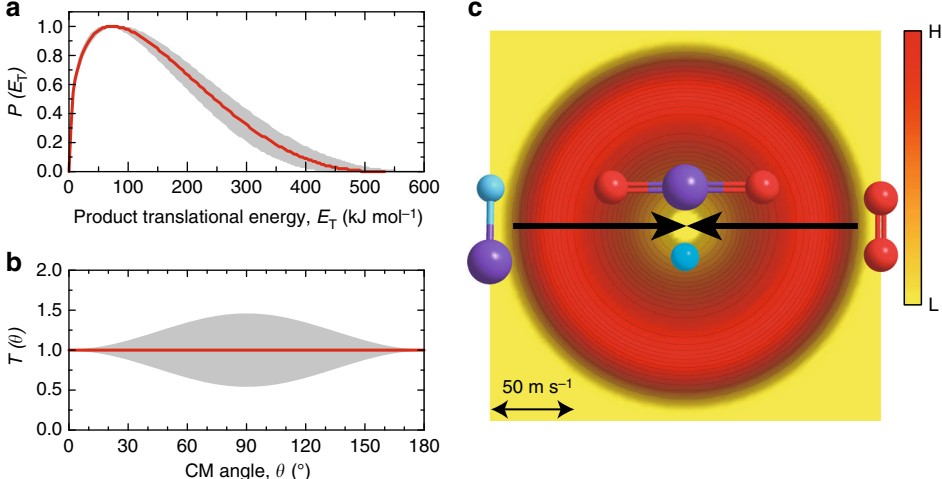

**Fig. 2** Center-of-Mass (CM) distributions and the associated flux contour map. CM translational energy flux distribution (**a**), CM angular flux distribution (**b**), and the top view of their corresponding flux contour map (**c**) leading to the formation of silicon dioxide plus atomic deuterium in the reaction of D1-silylidyne with molecular oxygen. Shaded areas indicate the error limits of the best fits accounting for the uncertainties of the laboratory angular distribution and TOF spectra, with the red solid lines defining the best-fit functions. The flux contour map represents the flux intensity of the reactive scattering products as a function of the CM scattering angle (θ) and product velocity (u). The color bar indicates the flux gradient from high (H) intensity to low (L) intensity. Colors of the atoms: silicon (purple), oxygen (red), and deuterium (light blue)

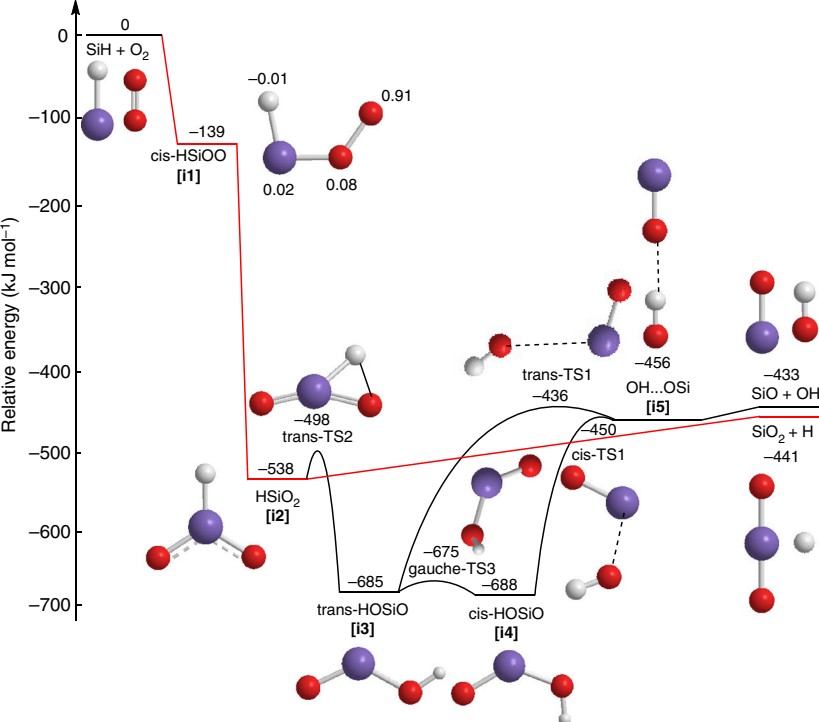

**Fig. 3** Potential energy surface. The potential energy surface for the reaction of the silylidyne radical with molecular oxygen including reaction pathways energetically accessible in the crossed molecular beam experiments. The route in red highlights the reaction pathway leading to the formation of silicon dioxide plus atomic hydrogen. Relative energies are given in units of kJ mol$^{-1}$. Note that the relative energy of silicon dioxide plus atomic deuterium is 3 kJ mol$^{-1}$ higher as compared to non-deuterated reactants, whereas for the intermediates and transition states in the SiD-O$_2$ and SiH-O$_2$ systems, relative energies are within 1 to 2 kJ mol$^{-1}$. For **[i1]**, the spin density distribution is also shown. Colors of the atoms: silicon (purple), oxygen (red), and hydrogen (light grey)

point vibrational energies by replacing hydrogen (H) with deuterium (D) in SiH, the relative energy of SiO$_2$ plus D is 3 kJ mol$^{-1}$ higher than that of non-deuterated reactants, whereas for the intermediates and transition states in the SiD-O$_2$ and SiH-O$_2$ systems, which maintain the Si-H(D) bond intact, relative energies are within 1 to 2 kJ mol$^{-1}$. The computations verify the experimental results of an indirect reaction mechanism. Here, the reaction is initiated by a barrierless addition of SiH with its radical center to O$_2$ at a single oxygen atom yielding a C$_s$ symmetric cis-HSiOO intermediate **[i1]** on the doublet surface. The barrierless addition was verified by a careful examination of the entrance channel, which indicates that the potential energy of the system monotonically decreases as SiH approaches O$_2$. The collision complex **[i1]** is only metastable and undergoes a rapid atomic oxygen migration to the silicon atom forming the C$_{2v}$ symmetric HSiO$_2$ ($^2B_1$) intermediate **[i2]**, which is strongly bound by 538 kJ mol$^{-1}$ with respect to the separated reactants. This intermediate can either undergo unimolecular decomposition via a loose exit transition state by H loss forming SiO$_2$ ($^1\Sigma_g^+$) or isomerize via hydrogen migration to trans-HOSiO (C$_s$, $^2$A', **[i3]**), which in turn undergoes trans-cis isomerization to cis-HOSiO (C$_s$, $^2$A', **[i4]**). Multireference CASPT2 calculations with full active spaces (17,13) corroborate the conclusions that the reversed addition reaction of H to SiO$_2$ is barrierless. The trans-HOSiO and cis-HOSiO intermediates are isovalent to the well-known trans-HOCO and cis-HOCO intermediates, reside in deep potential energy wells of 685 and 688 kJ mol$^{-1}$, and can undergo facile Si-O bond cleavages through loose exit transition states yielding a linear van-der-Waals complex between the hydroxyl radical (OH) and SiO **[i5]**, in which OH is hydrogen bridge bonded to the oxygen atom of SiO. This complex is bound by 23 kJ mol$^{-1}$ with respect to the separated products. Overall, the

computations revealed two competing exit channels: the formation of SiO$_2$ plus H and SiO plus OH. With the exception of **[i1]**, the aforementioned energetics are within 13 kJ mol$^{-1}$ when compared with Schaefer's study on the stationary points relevant to the reaction of SiO with OH;[39] Darling and Schlegel predicted the existence of **[i1]**, but their energetics, computed at the G2 level of theory, forecasted the energy difference between **[i1]** and **[i4]** to be about 505 kJ mol$^{-1}$ compared to 549 kJ mol$^{-1}$ in our system[40]. Finally, since our experimental setup could not probe the SiO route, the branching ratios were determined computationally exploiting Rice–Ramsperger–Kassel–Marcus (RRKM) theory (Methods). The relative yields of SiO$_2$ and SiO were virtually independent of the collision energy between 0 and 36 kJ mol$^{-1}$ and varied in the ranges of 49.5 ± 2.5 and 50.5 ± 2.5%. We should note, however, that the energy content in the intermediates **[i1]** to **[i5]** is so significant that the RRKM rate constants are close to the applicability of the statistical theory. Therefore, dynamical effects might affect the branching ratios, but a 50–50% partition is reasonable given the closeness of the reaction energies.

## Discussion
Let us first address the barrierless character of the reaction of SiH with O$_2$. The initial addition step to **[i1]** has no barrier because it represents an association of two species, each of them having at least one unpaired electron, in this case a radical (SiH) and a diradical (O$_2$). During the association process two unpaired electrons from the two interacting moieties form an electron pair thus creating a new Si-O single bond. Such radical/(di)radical reactions occur without barriers. While the complex **[i1]** is formed, the total spin of the doublet/triplet pair is converted to a doublet due to the formation of the extra electron pair. The

calculated spin density distribution in [i1] shows that the remaining unpaired electron in HSiOO is localized on the terminal oxygen atom, which exhibits a spin density of 0.91 (Fig. 3). It is interesting to note that the related reactions of Si with $O_2$ (diradical/diradical) and singly ionized silicon ($Si^+$) with $O_2$ (radical/diradical) are also barrierless[41,42]. However, these reactions cannot produce $SiO_2$ or singly ionized silicon dioxide ($SiO_2^+$) under single collision conditions as prevailing in the low density ISM. Here, $SiO_2$ and $SiO_2^+$ represent highly exothermic reaction intermediates residing in deep potential wells, and their internal energy due to chemical activation has to be dissipated. The energy dissipation may occur either via deactivation through third-body collisions or via fragmentation to SiO plus O or singly ionized silicon monoxide ($SiO^+$) plus O, respectively. Whereas third-body collisions prevent the dissociation of $SiO_2$ when the reaction of Si with $O_2$ takes place in a 0.37 K superfluid helium droplet[43], the fragmentation channels take over at low pressures where collision deactivations are inefficient. Indeed, the reaction of Si with $O_2$ is usually invoked as one of the main source of SiO[44]. Crossed molecular beam studies of this system have firmly established SiO plus O as the reaction products under nearly zero-pressure conditions[45] as corroborated by theoretical quasi-classical trajectory calculations[46]. An alternative reaction leading to the formation of $SiO_2$ plus O, i.e., the reaction of SiO with $O_2$, which plays an important role in fabrication of silicon oxide films at elevated temperatures[47], is known to be endothermic by 28 kJ $mol^{-1}$ and has a barrier of at least 91 kJ $mol^{-1}$ and hence cannot occur in low-temperature interstellar environments[48].

The SiH reaction with $O_2$ is akin to reaction of the isovalent methylidyne radical (CH) with $O_2$, which is also known to be barrierless and consequently very fast at the collisional kinetic limit even at extremely low temperatures such as 13 K[49]. Theoretically, low-temperature rate coefficients for this system have been evaluated using long-range transition state theory and the calculations reproduced the experimental values within a factor of two to three[50]. Here we exploited the same theoretical method to compute rate coefficients for the SiH-$O_2$ reaction and obtained values slightly increasing from $3.1 \times 10^{-10}$ $cm^3$ molecule$^{-1}$ s$^{-1}$ at 10 K to $4.4 \times 10^{-10}$ and $5.3 \times 10^{-10}$ $cm^3$ molecule$^{-1}$ s$^{-1}$ at 100 and 300 K, respectively, compared to an experimental value of $1.7 \times 10^{-10}$ $cm^3$ molecule$^{-1}$ s$^{-1}$ at 298 K[51]. A comparison of the rate coefficients at 13 K for the CH-$O_2$ ($1.5 \times 10^{-10}$ $cm^3$ molecule$^{-1}$ s$^{-1}$ (experiment) and $2.9 \times 10^{-10}$ $cm^3$ molecule$^{-1}$ s$^{-1}$ (theory)) and SiH-$O_2$ ($3.2 \times 10^{-10}$ $cm^3$ molecule$^{-1}$ s$^{-1}$ (theory)) reveals that both reactions should be nearly equally fast. The calculations also demonstrate that the long-range SiH/$O_2$ interaction is dominated by dispersion forces, with small contributions from dipole (SiH)—induced dipole ($O_2$) and dipole (SiH)—quadrupole ($O_2$) interactions.

Having established the formation of $SiO_2$ along with SiO under single collision conditions in the laboratory and through electronic structure calculations, we now discuss potential astrochemical implications. It is essential to transfer these findings to 'real' extraterrestrial environments since all experiments conducted under well-defined laboratory conditions can hardly mimic the chemical complexity of the ISM, where both neutral–neutral and ion–molecule reactions along with photochemical processes occur simultaneously[6,38,52]. Our studies indicate unambiguously that the reaction has no entrance barrier, all barriers involved in the formation of the silicon oxides are well below the energy of the separated reactants, and the overall reactions to form the silicon oxides are exothermic. These findings represent a crucial prerequisite for this reaction to be important in low-temperature molecular clouds; any barrier would block these reactions in low-temperature interstellar environments. Therefore, our results can be universally applied to any cold interstellar environment such as molecular clouds, where ice mantles can be removed from grains, and where adequate concentrations of SiH radicals and $O_2$ exist.

In cold molecular clouds, $O_2$ is difficult to detect. This is in part due to the fact that this diatomic molecule is homopolar, and that observations are best made from space platforms such as Herschel[53]. The former implies that large concentrations (or column densities) are needed for detection, the latter that small telescope diameters and large beam sizes dilute the signal intensity thus making molecular oxygen difficult to detect even in sources such as hot molecular cores. Hence, the recent detection of $O_2$ toward Orion and $\rho$ Ophiuchi A is truly extraordinary[53–56]. Gas phase abundances of $O_2$ with respect to molecular hydrogen ($H_2$) at the order of $10^{-7}$–$10^{-6}$ would be in good agreement with astronomical observations[53,57], however, they are still significantly lower than the upper limit to the $O_2$ abundance in interstellar ice. Due to its non-polar nature, upper limits to the abundance of $O_2$ in interstellar ice are difficult but its effect on the carbon monoxide (CO) vibrational absorption band at 4.673 μm (2140 $cm^{-1}$) is consistent with ice mixtures in which $O_2$ is comparable to or a few times more abundant than CO[57]. Vandenbussche et al. (1999) provided upper limits[58] on the $O_2$ ice abundance based on the non-detection of its fundamental vibrational band in the solid state, at 6.45 μm (1550 $cm^{-1}$), as well as on observations of the 4.673 μm (2140 $cm^{-1}$) CO band. Toward the protostar R CrA IRS2, they find an upper limit of 50 % with respect to solid CO and, in NGC7538 IRS9, an upper limit of 20 % to water ice which has an estimated abundance of $10^{-4}$ with respect to $H_2$. In general, the few sources in which observations have been carried out are consistent with upper limits to the $O_2$ ice abundances in the range $10^{-5}$–$10^{-4}$ with respect to $H_2$. This abundance is consistent with $O_2$ to water ratio (0.038) recently detected in comet 67 P/Churyumov–Gerasimenko[59]. Indeed, Taquet et al. argued that the high abundance of $O_2$ as seen in this 67 P is of interstellar origin[60]. Further, we should note that the abundance of SiH depends on the removal of the ice mantles. Removal provides a parent species (silane; $SiH_4$) to the gas phase with subsequent photodissociation producing SiH[28–30], a molecule with a very small dipole moment of only 0.12 Debye[61]. Gas phase detection has therefore been difficult with only a tentative identification to date in Orion[62]. While multiple reaction pathways can lead to $SiO_2$, the neutral–neutral reaction of SiH with $O_2$ has a universal potential to synthesize $SiO_2$ along with SiO in cold molecular clouds when the fractional abundance of $O_2$ in the gas phase is sufficient. Both $SiO_2$ and SiO—as derived from laboratory studies of barrier-less condensation of silicon oxides in helium droplet experiments[25,43] and theoretical chemistry calculations that show reactions between these oxides produce larger silicon oxides in exothermic chemistry[16,17,31]—may eventually play a role in the formation and growth of interstellar silicates. It should be noted that the elementary reactions leading to small silicon oxides may also be relevant to the chemistry of silicon oxide plasmas, such as those widely used in the semiconductor industry for depositing thin film insulators in integrated circuits like memory or processor chips[63]. During the Si and $SiO_2$ chemical vapor deposition (CVD) processes, primary precursors such as silane or disilane ($Si_2H_6$) undergo bond cleavage processes with the fragments adsorbing on the surfaces of the substrates, during which complex gas-phase and gas–surface reactions are involved[64]. Layered films of silicon oxides can be processed by mixing silane or disilane with $O_2$ and dinitrogen monoxide ($N_2O$), for instance[65]. Here depositions at low temperatures are preferred, since high-temperature instabilities on the substrates can substantially reduce the film production rates[66]. Plasma-enhanced CVD (PECVD) represents the main processing route for silicon oxide film growth; here, a radio

frequency (RF) discharge supplies energy to initiate bond rupture processes and hence to promote the chemical reaction[67,68]. Since silicon hydride radical species such as $SiH_m$ and $Si_2H_n$ ($m = 1$–$3$, $n = 1$–$5$) are generated as well, PECVD is extraordinarily complex[64,69]. Therefore, it is vital to untangle the reaction mechanisms from the fundamental, microscopic point of view.

Our combined crossed molecular beam and electronic structure calculations provide compelling evidence on the formation of $SiO_2$ along with $SiO$ under single collision conditions. The bimolecular neutral–neutral reaction of SiH with $O_2$ represents a single step mechanism to form two silicon oxides—$SiO$ and $SiO_2$. In combination with astrochemical modeling, our study suggests that silicon oxides may form not only in hot circumstellar envelopes of oxygen-rich stars as thought previously, but also in interstellar clouds via facile, barrierless reactions involving the simplest silicon-bearing radical (SiH) and $O_2$. These pathways provide a population of silicon oxides, which can possibly provide a basis for the regeneration of interstellar silicates thus leading us closer to solving the paradox of the injection and destruction timescales of silicates[5,21–24]. The ability of barrierless, exothermic reactions between $SiO$ and $SiO_2$ to form larger silicon oxides such as $Si_2O_5$ and $Si_3O_5$ suggests that such reactions[31] may play a central role in the process by which reformation of silicate grains must occur in the ISM.

Although there is no detailed description to date of how silicate dust grains might form in the ISM, silicon oxides are likely involved. Therefore, the work reported here represents an important step toward a systematic understanding of the fundamental chemical processes eventually leading to the formation of silicate grains in the ISM. Since distinct types of interstellar grains (silicates, carbonaceous grains, and silicon carbide) exist, the current work resembles a template for future studies of elementary chemical reactions relevant to grain formation. This requires a sophisticated link of laboratory, theoretical, and modeling study with astronomical observations, particularly for those high-density regions in circumstellar envelopes, hot molecular cores, shocked gas, including supernova remnants. Considering the crucial role of interstellar dust in star and hence solar system formation[70] and in the chemical evolution of the universe, with grains providing critical molecular factories to even synthesize bio-relevant organics like amino acids and sugars[7], the unraveling of the cosmic dust enigma is of fundamental importance to the understanding of our origins.

## Methods

**Crossed molecular beam experiments**. The experiments were conducted in a crossed molecular beam machine under single-collision conditions at the University of Hawai'i at Mānoa[71,72]. In the primary reactant chamber, a pulsed and supersonic beam of the D1-silylidyne radical (SiD; $X^2\Pi$) at fractions of about 0.5% was prepared in situ by ablation of a rotating silicon rod with the output from a Spectra-Physics Quanta-Ray Pro 270 Nd:YAG laser (30 Hz, 266 nm, 10–15 mJ pulse energies), with the ablated species further entrained by molecular deuterium ($D_2$, 99.7%; Icon Isotopes, Inc.). The $D_2$ gas acts as a carrier and reactant gas, and no other silicon-deuterium-bearing molecules were found to be present in the beam under the experimental conditions. Considering the natural isotope abundances of silicon, it was easier to optimize a SiD beam at $m/z$ 31. The molecular beam entraining the SiD radicals then passed a skimmer and a chopper wheel, generating a pulsed radical beam of a well-defined peak velocity of $1981 \pm 38\ \mathrm{m\ s^{-1}}$ and speed ratio of $5.2 \pm 1.0$. Notice that even if SiD radicals of $A^2\Delta$ state were formed, they can decay to the ground state during the travel time of about 18 μs to the interaction region of the scattering chamber[73]. In the scattering chamber, this segment crossed a supersonic beam of pure oxygen gas ($O_2$; 99.998%; Matheson) perpendicularly, which had a velocity of $778 \pm 20\ \mathrm{m\ s^{-1}}$ and a speed ratio of $15.6 \pm 1.0$. This setup eventually yielded a collision energy of $33.2 \pm 2.0\ \mathrm{kJ\ mol^{-1}}$ and a center-of-mass (CM) angle of $23.3 \pm 1.2°$. The reactive scattering products were ionized by an electron-impact ionizer operating at 80 eV and 2 mA emission current, before they entered a quadrupole mass spectrometer (QMS, Extrel QC 150) operating in the time-of-flight (TOF) mode. The selected ion species filtered by the QMS at a specific $m/z$ travelled towards a stainless steel target coated with a thin layer of aluminum biased at −22.5 kV and a cascade-of-electron pulse was

initiated upon impact. The electrons were then expelled from the stainless steel target and flew toward an organic scintillator to generate a photon pulse, prior to be detected by a Burle photomultiplier tube (PMT, Model 8850) operating at −1.35 kV. The signal was eventually filtered by a discriminator (Advanced Research Instruments, Model F-100TD) at a level of 1.6 mV before being fed into a Stanford Research System SR430 multichannel scaler. The whole detection region is housed in a triply, differentially pumped vacuum chamber, which can be rotated in a plane defined by the primary and secondary beams, thus we are able to record TOF spectra at discrete angles, integrate and normalize them with respect to the intensity at the CM angle, and then extract the product laboratory angular distribution at a specific $m/z$[74]. In order to obtain the information about the reaction dynamics, we employed a forward-convolution routine based on the Jacobian transformation to convert the data in the laboratory frame into the CM frame[32,75]. This method actually begins with a trial set of parameterized CM functions—the translational energy flux distribution, $P(E_T)$, and the angular flux distribution, $T(\theta)$, in the CM frame, to iteratively fit the laboratory TOF spectra and the angular distribution until the best fits are reached, accounting for apparatus performances, beam divergences, and velocity spreads. We can then plot a flux contour map, $I(\theta, u) = P(u) \times T(\theta)$, which presents the flux of the reactive scattering products as a function of the CM scattering angle ($\theta$) and product velocity ($u$), and reveals information on the scattering reaction dynamics[76].

**Electronic structure calculations**. Geometries of most of the intermediates, transition states, and bimolecular fragments were adapted from Schaefer et al.[39] where they were optimized at the CCSD(T)//cc-pV(Q + d)Z level of theory. Single-point energies were computed using the explicitly correlated CCSD(T)-F12 method[77] with the correlation-consistent aug-cc-pV5Z basis set[78,79]; this theoretical approach closely approximates CCSD(T) energies at the complete basis set (CBS) limit and is expected to provide relative energies with an accuracy of 5 kJ mol$^{-1}$. In addition to the structures considered by Schaefer et al.[39], we searched for cis-HSiOO and trans-HSiOO and cyclic $HSiO_2$ intermediates, which may represent initial complexes produced in the reaction of SiH with $O_2$. We employed the same CCSD(T)//cc-pV(Q + d)Z method for their geometry optimization. Only the cis-HSiOO structure [i1] was located as a local minimum on the potential energy surface. A trans-HSiOO structure spontaneously converges to the $HSiO_2$ isomer [i2] during optimization. A cyclic $HSiO_2$ geometry optimizes to a stationary point within $C_{2v}$ symmetry, but shows one imaginary frequency. Once symmetry is released, the further optimization yields the open $HSiO_2$ isomer [i2]. Next, we searched for a transition state connecting cis-HSiOO [i1] with $HSiO_2$ [i2]. The saddle-point optimization gave a non-planar structure typical for a transition state for rotation around the Si-O bond. The transition state would normally connect two planar cis-HSiOO and trans-HSiOO minima, but since the latter spontaneously rearranges to $HSiO_2$, this transition state is in fact between [i1] and [i2]. Furthermore, the transition state appears to be only slightly higher in energy than [i1] at the CCSD(T)//cc-pV(Q + d)Z level, and becomes lower in energy than [i1] when ZPE corrections are included. The transition state energy is below that of cis-HSiOO at our final CCSD(T)-F12/aug-cc-pV5Z + ZPE(CCSD(T)/cc-pV(Q + d)Z) level of theory. This result indicates that [i1] could be only a metastable structure on the potential energy surface, which rapidly isomerizes to $HSiO_2$ [i2]. In this view, the entrance channel of the SiH + $O_2$ reaction can be described as addition of $O_2$ to the radical site on the Si atom either in cis-conformation or trans-conformation followed by spontaneous rearrangement to $HSiO_2$ resulting in formal insertion of HSi into the O = O bond of $O_2$. The process is driven by its very-high exothermic energy of 538 kJ mol$^{-1}$. We have also carefully mapped out the potential energy surface for H loss from $HSiO_2$ [i2] to verify that this process occurs without an exit barrier. Within $C_{2v}$ symmetry, the electronic state of [i2] is $^2B_2$ but the products, $SiO_2$ + H, correlate to the $^2A_1$ state. Therefore, the H loss is a symmetry-forbidden process within the $C_{2v}$ point group. If symmetry is reduced to $C_s$, the two states involved are both $^2A'$ and hence H elimination should take place via a crossing or avoided crossing of the two $^2A'$ states. Therefore, we scanned the minimal energy path for the H loss from [i2] using the multireference second-order perturbation theory CASPT2 method[80] with the full valence active space containing 17 electrons distributed on 13 orbitals (17,13) with the aug-cc-pVDZ basis set. The CASPT2(17,13)/aug-cc-pVDZ geometry optimizations and vibrational frequency calculations were carried out for various Si-H distances frozen at the values from 1.5 Å to 4.0 Å with a step of 0.2 Å. Following these geometry optimizations, relative energies of the MEP structures were refined by employing CASPT2(17,13) calculations with a larger aug-cc-pVTZ basis set and with mixing wave functions of the two lowest $^2A'$ involved in the process. The MEP has shown a steady energy decrease with the Si-H distance from [i2] to the $SiO_2$ + H products confirming the barrierless character of the H elimination. All CCSD(T), CCSD(T)-F12, and CASPT2 calculations in this work were performed using the MOLPRO 2010 program package[81]. Collision-energy dependent rate constants for chemically activated unimolecular reactions on the $HSiO_2$ surface starting from the intermediate [i2] were carried out employing RRKM theory using the in-house code[82]. The internal energy of each intermediate was taken as a sum of its chemical activation energy (the negative of its relative energy with respect to the SiH + $O_2$ reactants) plus the collision energy. For reaction steps proceeding well-defined transition states, we used CCSD(T)-F12/aug-cc-pV5Z relative energies calculated here and CCSD(T)/cc-pV(Q + d)Z vibrational frequencies reported by Schaefer

et al.[39] For the [i2] → $SiO_2$ + H barrierless reaction step, we employed variational transition state theory (VTST)[83,84] with transition state candidate structures ranging along the MEP. Here we used vibrational frequencies of the MEP structures calculated at the CASPT2(17,13)/aug-cc-pVDZ level and their relative energies obtained at CASPT2(mix = 2,17,13)/aug-cc-pVTZ. For barrierless dissociation of the OH…OSi complex [i5] to the SiO + OH products, we also utilized the VTST approach. Here, the wave function is single-reference, the interaction has a van der Waals character, and hence the MEP was mapped out using a density functional wB97XD method[85] including a dispersion correction with the aug-cc-pVTZ basis set. Single-point relative energies of the MEP structures were refined at our best CCSD(T)-F12/aug-cc-pV5Z level and were used in VTST calculations in conjunction with wB97XD/aug-cc-pVTZ frequencies. The dipole and quadrupole moments and polarizabilities of SiH and $O_2$ required for the calculations of the capture rate coefficients using long-range transition state theory were also computed at the wB97XD/aug-cc-pVTZ level of theory. The unimolecular rate constants obtained from the RRKM and VTST calculations for collision energies in the 0–33.2 kJ mol$^{-1}$ interval were used to compute the branching ratios for the formation of the $SiO_2$ + H and SiO + OH products within the reaction scheme illustrated in Fig. 3 employing the steady-state approximation. It should be noted that because of very-high-energy content in the intermediate [i2] and relatively low activation energies required for isomerization and dissociation steps, the product branching ratios appeared to be insensitive to the collision energy within the considered range.

**Data availability**. The data that support the findings of the current research are available from the corresponding authors upon request.

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

## Acknowledgements

We thank the National Science Foundation (NSF) for support under award CHE-1360658. Astrophysics at Queen's University Belfast is supported by a grant, ST/P000321/1, from the Science and Technology Facilities Council (UK).

## Author contributions

T.Y., A.M.T., and B.B.D. carried out the experimental measurements; T.Y. performed the data analysis; A.M.M. carried out the theoretical analysis; T.J.M. carried out the astronomical discussion; T.Y., R.I.K. and A.M.M. wrote the paper; and T.Y. and R.I.K. supervised the study.

## Additional information

**Competing interests:** The authors declare no competing financial interests.

