## [Peer Review File · Nature Communications]

Reviewers' comments:

Reviewer #1 (Remarks to the Author):

Directed Gas-Phase Formation of Silicon Dioxide and Implications to the Formation of Interstellar Silica.

This manuscript reports a combined crossed molecular beam and theoretical study which finds that the molecule SiO₂ may be formed through the reaction of SiH and O₂ with a negligible barrier. This is hypothesized to be a mechanism by which SiO₂ may form at low temperatures including in the Interstellar Medium. As the question of how silicates found in the interstellar medium are formed, it is an interesting paper, that addresses a significant problem in astrochemistry.

The fact that the O₂ cleavage is barrierless is somewhat surprising. The paper would be much improved if this result was placed in better context. How does this reaction compare to other similar reactions such as Si+O₂, Si(+)+O₂, SiO + O₂ and other reactions. It is not clear why SiH forms SiO₂ while say Si+O₂ would not. There is little effort to explain why this reaction is barrierless. Also, the role of spin should be expanded upon. Is the spin of O₂ quenched in i1 complex? Theory tends to have difficulty with O₂ due to delocalization error removing spin from O₂.

The question of whether SiO formation is occurring is an important one. Can Si²⁹ or Si³⁰ be used to observe the SiO formation?

How does this result apply to the formation of Silicate clusters and nanoparticles? If SiO₂ may be formed in the interstellar medium, the problem of forming interstellar grains remain. There is a large amount of O₂ in the ISM, then this becomes a more likely possibility. How does this result generalize to the formation of larger particles.

A major revision is necessary for me to recommend accepting this manuscript, with the major issue being explaining why the reaction has such a low barrier, demonstrating that this specific reaction is critical for SiO₂ formation at low temperature and showing that this may lead to the formation of grains.

Reviewer #2 (Remarks to the Author):

The Authors report an experimental study of the reaction between the D1-silyldiyne (SiD) and dioxygen (O₂) species in the gas phase as a proxy to the reaction between silyldiyne (SiH) and O₂. Combining the experimental measurements with the results of theoretical calculations on the SiH + O₂ reaction, they demonstrate convincingly that the reaction of SiH with O₂ does not require activation energy and thus can take place at very low temperature. The Authors have found that the reaction has two possible results with the same probability of 50%, on the one hand, silicon dioxide plus hydrogen (SiO₂ + H), and, on the other hand, silicon monoxide plus hydroxyl (SiO + OH).

This original study is of interest to chemistry in general, more specifically to chemistry in the gas phase at very low temperature. Its principal application concerns the modelling of interstellar chemistry. As the SiH + O₂ reaction is currently not taken into account in the interstellar chemical reaction network, the present results will contribute to the development of this network. As stated by the Authors, the description of the life cycle of cosmic silicate grains is not complete as a growth mechanism of the grains in the interstellar medium appears to be missing. The reaction demonstrated in this study is a potential source of interstellar SiO₂ and SiO molecules, which may play a role in this mechanism.

I recommend the publication of this manuscript provided that the following issues are taken care of.

My only major concern is the lack of direct relevance to the text that affects the references in the introductory part, and in the Discussion and Conclusions sections:

- (lines 57-60) "Current astrochemical models of circumstellar envelopes propose that dust formation in AGB stars is driven eventually by clustering and reactions of silicon oxides along with magnesium- iron-type oxides^{2,4,14-22}": only two of the articles used as references actually deal with astrochemical models of circumstellar envelopes. The others do not concern circumstellar chemistry. (I could not check reference 4.) Other relevant studies may be available.
- (line 64-65) "... , it remains possible that significant formation of dust needs to occur in the interstellar as opposed to the circumstellar medium^{11-14,26,27}": references 26 and 27 are experimental studies on the formation of dust grains; these studies do not examine the need for the interstellar formation of grains, but they regard its possibility as a motivation. I propose to add a sentence at the end of the paragraph: "... as opposed to the circumstellar medium¹¹⁻¹⁴. Actually, silicate grains may grow in the ISM by accreting and incorporating silicon oxide molecules^{26,27}.
- (lines 203-205) "Both silicon dioxide and silicon monoxide – ... – may eventually play a role in the formation of interstellar silicates^{51,52}": given the context, I assume "formation" stands here for growth in the interstellar medium; references 51 and 52, however, concern the formation of silicates in circumstellar environments.
- (lines 216-219) "These pathways provide a population of silicon oxides, which can possibly provide a basis for the regeneration of interstellar silicates thus leading us closer to solving the paradox of the injection and destruction timescales of interstellar silicates^{21,53}": "these pathways" denotes the reaction between SiH and O₂ mentioned in the sentence that precedes this statement; references 21 and 53 are articles on the relation between the abundance of SiO and shocks in two types of interstellar regions.
- (lines 230-231) "Considering the crucial role of interstellar dust in star and hence solar system formation²⁶": reference 26 is essentially an experimental study on the formation of silicon oxide grains at low temperature and does not deal with the role of dust in the formation of stars.

I would also make the following suggestions with regard to references:

- (line 62) "that limits their lifetime to only a few 10⁸ years^{5,23-25}": add Draine (2009) as a reference for lifetimes ("Interstellar Dust Models and Evolutionary Implications"; link: <http://aspbooks.org/custom/publications/paper/414-0453.html>).
- (lines 171-175) "It is essential to transfer these findings to 'real' extraterrestrial environments since all experiments conducted under well-defined laboratory conditions can hardly mimic the chemical complexity of the interstellar medium, where both neutral-neutral and ion-molecule reactions along with photochemical processes occur simultaneously^{39,42}": references 39 and 42 are articles reporting each on a specific species and the related chemistry; the addition of a reference to a more general article on interstellar chemistry is desirable.
- (lines 189-190) "Hence, the recent detection of molecular oxygen toward Orion and ρ Oph A is truly extraordinary^{43,44}": add Goldsmith et al. (2011) for anteriority (link: <http://dx.doi.org/10.1088/0004-637X/737/2/96>).

In the experimental part, the reference for normalization of the counts should be mentioned:

- (lines 265-266) "we are able to record TOF spectra ... and normalize them, ..."
- (line 358, in the title of the vertical axis, in Figures 1A and 1B) "Normalized counts"

The following statements could be made clearer:

- (lines 47-48) "Interstellar silicates [act] as a molecular feedstock³": I do not see an explicit connection with reference 3.
- (lines 190-192) "[abundances] higher than ... are significantly lower than ...": this is not logical.

Suggested minor corrections to the text:

- (line 51) "the formamide": formamide
- (line 55) "via the late stages": during the late stages
- (line 116) "photo detachment": electron photodetachment
- (line 153) "the OH": OH
- (lines 154-155) "the OH": OH
- (line 181) "such as to molecular clouds": such as molecular clouds
- (line 258) "running at -22.5 kV": biased with -22.5 kV
- (line 258) "a cascade of electron pulse was initiated": a cascade of electrons was initiated upon impact / a cascade-of-electrons pulse was initiated upon impact

Reviewer #3 (Remarks to the Author):

This is an excellent paper that shows conclusively that silicon monoxide and silicon dioxide can both be formed from the gas phase bimolecular reaction of the SiH radical and molecular oxygen without any activation energy, and therefore at the very low temperatures characteristic of the interstellar medium. This is an important new result, and I am persuaded by the authors' arguments that it suggests a significant possible new source of silicates in the interstellar medium and possibly moves the production/destruction ratio of silicates in the ISM in a good direction. The molecular beam apparatus and the methods applied are capable of producing fairly definitive evidence of what is claimed. The computational chemistry is done at a very high and believable level and is completely supportive of the experimental beam results. The astrophysical analysis of the relevance to the ISM seems solidly and understandably presented to me. I do have a few somewhat minor suggestions or comments.

1. The authors do not comment on whether their experiments can lead to any estimate of the rate constant for these reactions, or in other words on the cross-sections for reactive scattering. Even though the branching ratios they do report are themselves interesting, estimates of the absolute cross-sections would be required to incorporate them into the actual models of the chemistry and physics of the ISM.

2. The word exoergic has become somewhat ambiguous, because some authors, in fact some whole fields, use it to connote a negative change in the Gibbs free energy rather than in the enthalpy or internal energy. Therefore I myself prefer the older term exothermic, which is unambiguous in this regard.

3. The authors do not mention it, but I suspect that their result would also be quite significant to the modeling of laboratory silicon oxide plasmas, such as those widely used in the semiconductor industry for depositing thin film insulators in integrated circuits like memory or processor chips. This would again be especially true if reactive scattering cross-sections became available.

4. There are a few sentences where a word is missing, or not the best choice of word is used. This needs to be cleaned up a bit.

Reviewer 1

This manuscript reports a combined crossed molecular beam and theoretical study which finds that the molecule SiO₂ may be formed through the reaction of SiH and O₂ with a negligible barrier. This is hypothesized to be a mechanism by which SiO₂ may form at low temperatures including in the Interstellar Medium. As the question of how silicates found in the interstellar medium are formed, it is an interesting paper, that addresses a significant problem in astrochemistry.

The fact that the O₂ cleavage is barrierless is somewhat surprising. The paper would be much improved if this result was placed in better context. How does this reaction compare to other similar reactions such as Si+O₂, Si(+)+O₂, SiO + O₂. It is not clear why SiH forms SiO₂ while say Si+O₂ would not. There is little effort to explain why this reaction is barrierless.

We included a section describing why the radical - diradical reaction of SiH with O₂ has no barrier in the beginning of the Discussion section. Here, we also compared SiH + O₂ with the barrierless reactions Si + O₂ and Si⁺ + O₂ and explained why those reactions do not form SiO₂ and SiO₂⁺, respectively, in low-pressure environments. The SiO + O₂ reaction is also considered and we noted that although this reaction can produce SiO₂, it cannot occur at low temperature due to a high activation energy (barrier). In addition, we compared SiH + O₂ with the isoelectronic CH + O₂ reaction. In the revised version, we discussed the role of spin in the SiH + O₂ → [i1] association step.

Also, the role of spin should be expanded upon. Is the spin of O₂ quenched in i1 complex? Theory tends to have difficulty with O₂ due to delocalization error removing spin from O₂.

We cannot agree with the point that “Theory tends to have difficulty with O₂ due to delocalization error removing spin from O₂.” Because we used the state-of-the-art CCSD(T)-F12 approach in our calculations capable of reproducing of the energetics for the whole variety of chemical processes including the doublet + triplet → doublet association with accuracy of 5 kJ mol⁻¹ or better.

The question of whether SiO formation is occurring is an important one. Can Si²⁹ or Si³⁰ be used to observe the SiO formation?

²⁹SiO and ³⁰SiO would show up at m/z = 45 and 46, respectively. This overlaps with ¹³CO₂ as present as a residual gas in the detector. Most important, the low abundance of ²⁹Si and ³⁰Si delimits a detection of these species in our experiments. However, we exploited electronic structure calculations to demonstrate explicitly that the SiO channel is open with branching ratios of about 50 %.

How does this result apply to the formation of Silicate clusters and nanoparticles? If SiO₂ may be formed in the interstellar medium, the problem of forming interstellar grains remain. There is a large amount of O₂ in the ISM, then this becomes a more likely possibility. How does this result generalize to the formation of larger particles.

We modified the manuscript in multiple sections. For example, we included 'and growth' in abstract to note that molecules can also accrete on pre-existing or surviving silicate particles. Most important, just before the conclusions section, we added text to refer to the Avramov paper documenting the generalized principle that larger oxides can be built up in exoergic reactions.

A major revision is necessary for me to recommend accepting this manuscript, with the major issue being explaining why the reaction has such a low barrier, demonstrating that this specific reaction is critical for SiO₂ formation at low temperature and showing that this may lead to the formation of grains.

We addressed these concerns in the previous comments.

Reviewer 2

The Authors report an experimental study of the reaction between the D1-silyldiyne (SiD) and dioxygen (O₂) species in the gas phase as a proxy to the reaction between silyldiyne (SiH) and O₂. Combining the experimental measurements with the results of theoretical calculations on the SiH + O₂ reaction, they demonstrate convincingly that the reaction of SiH with O₂ does not require activation energy and thus can take place at very low temperature. The Authors have found that the reaction has two possible results with the same probability of 50%, on the one hand, silicon dioxide plus hydrogen (SiO₂ + H), and, on the other hand, silicon monoxide plus hydroxyl (SiO + OH).

This original study is of interest to chemistry in general, more specifically to chemistry in the gas phase at very low temperature. Its principal application concerns the modelling of interstellar chemistry. As the SiH + O₂ reaction is currently not taken into account in the interstellar chemical reaction network, the present results will contribute to the development of this network. As stated by the Authors, the description of the life cycle of cosmic silicate grains is not complete as a growth mechanism of the grains in the interstellar medium appears to be missing. The reaction demonstrated in this study is a potential source of interstellar SiO₂ and SiO molecules, which may play a role in this mechanism.

I recommend the publication of this manuscript provided that the following issues are taken care

of.

Thank you.

My only major concern is the lack of direct relevance to the text that affects the references in the introductory part, and in the Discussion and Conclusions sections:

(lines 57-60) "Current astrochemical models of circumstellar envelopes propose that dust formation in AGB stars is driven eventually by clustering and reactions of silicon oxides along with magnesium- iron-type oxides^{2,4,14-22}"; only two of the articles used as references actually deal with astrochemical models of circumstellar envelopes. The others do not concern circumstellar chemistry. (I could not check reference 4.) Other relevant studies may be available.

We fixed this.

- (line 64-65) "..., it remains possible that significant formation of dust needs to occur in the interstellar as opposed to the circumstellar medium^{11-14,26,27}"; references 26 and 27 are experimental studies on the formation of dust grains; these studies do not examine the need for the interstellar formation of grains, but they regard its possibility as a motivation. I propose to add a sentence at the end of the paragraph: "... as opposed to the circumstellar medium¹¹⁻¹⁴. Actually, silicate grains may grow in the ISM by accreting and incorporating silicon oxide molecules^{26,27}."

We changed this as well.

- (lines 203-205) "Both silicon dioxide and silicon monoxide – ... – may eventually play a role in the formation of interstellar silicates^{51,52}"; given the context, I assume "formation" stands here for growth in the interstellar medium; references 51 and 52, however, concern the formation of silicates in circumstellar environments.

We changed the formulation.

- (lines 216-219) "These pathways provide a population of silicon oxides, which can possibly provide a basis for the regeneration of interstellar silicates thus leading us closer to solving the paradox of the injection and destruction timescales of interstellar silicates^{21,53}"; "these pathways" denotes the reaction between SiH and O₂ mentioned in the sentence that precedes this statement;

references 21 and 53 are articles on the relation between the abundance of SiO and shocks in two types of interstellar regions.

This has been taken care of.

- (lines 230-231) "Considering the crucial role of interstellar dust in star and hence solar system formation²⁶": reference 26 is essentially an experimental study on the formation of silicon oxide grains at low temperature and does not deal with the role of dust in the formation of stars.

This has been fixed.

- (line 62) "that limits their lifetime to only a few 108 years^{5,23-25}": add Draine (2009) as a reference for lifetimes ("Interstellar Dust Models and Evolutionary Implications"; link: <http://aspbooks.org/custom/publications/paper/414-0453.html>).

We added this.

- (lines 171-175) "It is essential to transfer these findings to 'real' extraterrestrial environments since all experiments conducted under well-defined laboratory conditions can hardly mimic the chemical complexity of the interstellar medium, where both neutral-neutral and ion-molecule reactions along with photochemical processes occur simultaneously^{39,42}": references 39 and 42 are articles reporting each on a specific species and the related chemistry; the addition of a reference to a more general article on interstellar chemistry is desirable.

We added a reference.

- (lines 189-190) "Hence, the recent detection of molecular oxygen toward Orion and ρ Oph A is truly extraordinary^{43,44}": add Goldsmith et al. (2011) for anteriority (link: <http://dx.doi.org/10.1088/0004-637X/737/2/96>).

This has been added.

In the experimental part, the reference for normalization of the counts should be mentioned:

- (lines 265-266) "we are able to record TOF spectra ... and normalize them, ..."

- (line 358, in the title of the vertical axis, in Figures 1A and 1B) "Normalized counts"

We added the reference.

The following statements could be made clearer:

- (lines 47-48) "Interstellar silicates [act] as a molecular feedstock³": I do not see an explicit connection with reference 3.

We changed the reference.

- (lines 190-192) "[abundances] higher than ... are significantly lower than ...": this is not logical.

We corrected it.

Suggested minor corrections to the text:

- (line 51) "the formamide": formamide

- (line 55) "via the late stages": during the late stages

- (line 116) "photo detachment": electron photodetachment

- (line 153) "the OH": OH

- (lines 154-155) "the OH": OH

- (line 181) "such as to molecular clouds": such as molecular clouds

- (line 258) "running at -22.5 kV": biased with -22.5 kV

- (line 258) "a cascade of electron pulse was initiated": a cascade of electrons was initiated upon impact / a cascade-of-electrons pulse was initiated upon impact

We changed all of them, too. Thank you.

Reviewer 3

This is an excellent paper that shows conclusively that silicon monoxide and silicon dioxide can both be formed from the gas phase bimolecular reaction of the SiH radical and molecular oxygen without any activation energy, and therefore at the very low temperatures characteristic of the interstellar medium. This is an important new result, and I am persuaded by the authors' arguments that it suggests a significant possible new source of silicates in the interstellar medium and possibly moves the production/destruction ratio of silicates in the ISM in a good direction. The molecular beam apparatus and the methods applied are capable of producing fairly definitive evidence of what is claimed. The computational chemistry is done at a very high and believable level and is completely supportive of the experimental beam results. The astrophysical analysis of the relevance to the ISM seems solidly and understandably presented to me.

Thank you!

I do have a few somewhat minor suggestions or comments.

The authors do not comment on whether their experiments can lead to any estimate of the rate constant for these reactions, or in other words on the cross-sections for reactive scattering. Even though the branching ratios they do report are themselves interesting, estimates of the absolute

cross-sections would be required to incorporate them into the actual models of the chemistry and physics of the ISM.

In the revised version, we estimated rate coefficients for $\text{SiH} + \text{O}_2$ using long-range transition state theory and included the corresponding discussion and comparison with the related $\text{CH} + \text{O}_2$ reaction. We also added one reference (Nemoto et al.) which measured the rare constant albeit at 298 K.

2. The word exoergic has become somewhat ambiguous, because some authors, in fact some whole fields, use it to connote a negative change in the Gibbs free energy rather than in the enthalpy or internal energy. Therefore I myself prefer the older term exothermic, which is unambiguous in this regard.

We changed exoergic to exothermic throughout the manuscript.

3. The authors do not mention it, but I suspect that their result would also be quite significant to the modeling of laboratory silicon oxide plasmas, such as those widely used in the semiconductor industry for depositing thin film insulators in integrated circuits like memory or processor chips. This would again be especially true if reactive scattering cross-sections became available.

We added one section in the manuscript. Thank you for pointing this out.

4. There are a few sentences where a word is missing, or not the best choice of word is used. This needs to be cleaned up a bit.

We fixed these, too. Thank you.

Reviewers' comments:

Reviewer #1 (Remarks to the Author):

The paper is pretty interesting, and worthy of publication, however I'm not convinced that their arguments will be persuasive to the scientific community.

The authors find that all related O₂-radical reactions are barrierless. The importance of the H in SiH is that it can carry off energy. Since most grain formation will require the formation of larger clusters, collisions between O₂ and the cluster/grains can directly lead to grain growth and are likely to be large enough to absorb the excess energy. If barrierless O₂ reactions with silicates are common, this seems to diminish the suggested importance of SiH in grain formation. Their explanation seems in line with a hydrogen evaporation model for grain formation.

The authors could not experimentally demonstrate the branching ratios for SiO/SiO₂. They argue from theory only that it is 50/50. Experimental evidence would be significantly more convincing.

I thought much of the challenge with silicate formation is that lack of available oxygen. If we assume abundant O₂, then that seems to take care of the problem. The discussion of the astrochemical implications was interesting, but not particularly illuminating.

Lastly, I'd like to note that I had a question was about the whether the spin of the doublet HSiO₂ was localized on the O₂ complex or not. I'm not certain where the author's were going with their response, but it would still be nice to know the location of the spin density on the i1 complex.

Reviewer #2 (Remarks to the Author):

Dear Authors, dear Editors,

My appreciation of the manuscript has not changed and I still recommend its publication, provided that the remarks below are taken into account. I especially advise the authors to be more rigorous in their choice of references.

- (lines 47-48) "Interstellar silicates also play a critical role in star formation and in the origin of solar systems contributing to the radiation balance and acting as a molecular feedstock⁷": the review article proposed as a reference concerns organic molecules in astrophysics. It does not provide the reader with information on the role of silicates in star formation and in the origin of solar systems, etc., even though it mentions silicate grains as components of cosmic dust and comets. Please refer the reader to relevant literature and clarify the meaning of "Interstellar silicates [act] as a molecular feedstock³". Are silicates decomposed into molecules? Or do they adsorb and then release molecules? What molecules are concerned? A feedstock for what process?

- (lines 190-191) "At low pressures where collision deactivations are inefficient, the fragmentation channels take over": it would be good to give the current reference 61 as an example of third-body collisions, as in "Whereas third-body collisions prevent the dissociation of SiO₂ when the reaction Si + O₂ takes place in a 0.37-K superfluid He droplet⁶¹, the fragmentation channels take over at low pressures where collision deactivations are inefficient."

- (line 193) "studies ... has firmly": studies ... have firmly

- (lines 197-198) "... i.e. the reaction of silicon monoxide (SiO) with molecular oxygen

(O_2), which plays an important role in fabrication of silicon oxide films at elevated temperatures^{46,47}: one would expect references 46 and 47 to describe an SiO_2 -film production process that (i) involves the neutral-neutral $SiO + O_2$ reaction, (ii) requires a high temperature environment (a heated substrate, I guess), and (iii) is widely used, making the reaction important. I do not see how reference 47 is relevant, and I cannot verify reference 46 except for its abstract. In the latter case, the following document is more accessible and likely equivalent: Ishikawa, Matsugatani, and Takaoka, *Vacuum* 39 (11–12), 1111–1113 (1989) [link: [https://doi.org/10.1016/0042-207X\(89\)91101-9](https://doi.org/10.1016/0042-207X(89)91101-9)]. The SiO_2 -film production method it describes, which I expect to be the same as in reference 46, actually uses charged reactants, SiO^+ and O_2^+ (from an ICB source and a microwave ion source, respectively), not neutral species. It is also introduced as a method to produce good quality films while avoiding high temperatures. Therefore reference 46 is likely not relevant. Please refer the reader to literature on the role of the $SiO + O_2$ reaction in the fabrication of silicon oxide films at elevated temperatures.

- (lines 236-237) "Hence, the recent detection of molecular oxygen toward Orion and ρ Oph A is truly extraordinary⁵³⁻⁵⁵": add Larsson et al. (2007) for the first certain observation of O_2 in ρ Oph A [link: <http://dx.doi.org/10.1051/0004-6361:20065500>].

- (line 237) "Gas phase abundances of molecular oxygen higher than 10^{-6} would be ...": to be more precise, since reference 53 gives fractional abundances (relative to H_2) please write "Gas-phase abundances of molecular oxygen higher than 10^{-6} with respect to H_2 would be ..."

- (lines 237-239) "[abundances] higher than ... are significantly lower than ...": this is not logical. Here is an example with numbers: stating "abundances higher than 10 are in conflict with observations, but they are significantly lower than 100" does not make sense, because an abundance of 200 is higher than 10 and not lower than 100. Please rephrase the statement into "abundances in the range ... are higher than ..., but lower than ...", or "abundances of the order of ... are higher than ..., but lower than ...", for instance, with explicit values.

- (line 239) "... molecular oxygen abundance in interstellar ice⁵⁶": unless I am wrong, reference 56 does not give any definitive information on the abundance of molecular oxygen in interstellar ices and finally interprets observations assuming pure CO ices. At any rate, for a valid comparison, please refer to an article that gives the abundance of O_2 in interstellar ice with respect to that of H_2 , as done for gas-phase O_2 in reference 53, or derive it explicitly from reference 56 so as to make your point clear to the reader. Maybe cite Vandebussche et al. (1999) [link: <http://adsabs.harvard.edu/full/1999A%26A...346L..57V>] who give abundances of O_2 in the gas-phase and in ice relative to H (not H_2). Note that the abundance of O_2 in ice is not even an order of magnitude larger than its abundance in the gas phase according to these authors, unless the entire oxygen deficit is assigned to O_2 ice; it is more likely contained in H_2O ice, however. Note that the recent search for O_2 by Wirström et al. (2016) [link: <https://doi.org/10.3847/0004-637X/830/2/102>] did not detect the molecule although the targeted regions were expected to be favorable. Upper limits for the fractional abundance of O_2 consistent with previous studies were determined ($N(O_2)/N(H_2) = (0.6-1.6) \times 10^{-7}$).

- (lines 250-254) "Both silicon dioxide and silicon monoxide – ... produce larger silicon oxides in exothermic chemistry³¹ – may eventually play a role in the formation and growth of interstellar silicates^{16,17}": according to their authors, references 16 and 17 concern the formation of silicates in circumstellar environments (of course it could be extended to the growth of silicates in the ISM, but the authors did not do it explicitly). These references must be

inserted earlier in the sentence alongside reference 31: "Both silicon dioxide and silicon monoxide – ... produce larger silicon oxides in exothermic chemistry^{16,17,31} – may eventually play a role in the formation and growth of interstellar silicates." To propose references at the end of the sentence may not be necessary given that the formation (in the ISM) and especially the growth of interstellar silicates are phenomena often raised and speculated upon, but insufficiently studied experimentally to date.

- (lines 256-258) "[silicon oxide plasmas] widely used in the semiconductor industry for depositing thin film insulators in integrated circuits like memory or processor chips⁶²⁻⁶⁴": references 63 and 64 deal with the chemical vapor deposition of diamond, which is not related to silicon oxide plasmas or the semiconductor industry (not yet at any rate). Please provide relevant references.

- (lines 280-283) "These pathways provide a population of silicon oxides, which can possibly provide a basis for the regeneration of interstellar silicates thus leading us closer to solving the paradox of the injection and destruction timescales of silicates through interstellar shocks^{71,72}": simply revert to the original sentence (without "through interstellar shocks") and use the references given in the introduction as in "... the paradox of the injection and destruction timescales of silicates^{5,21-24}."

In the experimental part, the reference for normalization of the counts (i.e., what the counts are normalized against) should be mentioned:

- (lines 329-330) "we are able to record TOF spectra ... and normalize them, ..."

- (line 425, in the title of the vertical axis, in Figures 1A and 1B) "Normalized counts"

- (line 491) in reference 18, add the missing initials: Plane, J. M. C. & Bromley, S. T.

- (line 581) in reference 64, correct the name of the journal: "J. Phys.: Condens. Matter" (with colon and Matter, not "Mat.").

Reviewer #3 (Remarks to the Author):

I believe the manuscript can now be published in its present form. The authors' responses to all the suggestions made by the referees seem to be good ones and have led to a significantly improved paper. In particular the responses to my own suggestions seem to be completely satisfactory.

Reviewer #1 (Remarks to the Author):

The paper is pretty interesting, and worthy of publication, however I'm not convinced that their arguments will be persuasive to the scientific community.

The authors find that all related O₂-radical reactions are barrierless. The importance of the H in SiH is that it can carry off energy.

This is correct.

Since most grain formation will require the formation of larger clusters, collisions between O₂ and the cluster/grains can directly lead to grain growth and are likely to be large enough to absorb the excess energy.

There is no evidence supporting this statement ('likely'). Also, at the ultralow temperatures of 10 K in the ISM, O₂ would simply 'stick' to the grain and form – together with water – an icy mantle.

If barrierless O₂ reactions with silicates are common, this seems to diminish the suggested importance of SiH in grain formation. Their explanation seems in line with a hydrogen evaporation model for grain formation.

We did not state that 'barrierless O₂ reactions with silicates are common'. Silicates are closed shell species, and there are barriers between O₂ and closed shell species, but not with radicals. Also, there is no 'hydrogen evaporation' in the ISM – this is a sublimation; evaporation infers that hydrogen is in the liquid phase, which is incorrect.

The authors could not experimentally demonstrate the branching ratios for SiO/SiO₂. They argue from theory only that it is 50/50. Experimental evidence would be significantly more convincing.

Not a single experiment can answer all open experimental questions. Therefore, we can only present evidence from theory that the branching ratio is 50/50.

I thought much of the challenge with silicate formation is that lack of available oxygen. If we assume abundant O₂, then that seems to take care of the problem.

We do not assume abundant O₂. We are citing results from observations and laboratory studies.

The discussion of the astrochemical implications was interesting, but not particularly illuminating.

Referee 1 did not present specific suggestions/requests how to change the implications, therefore no modification could be carried out.

Lastly, I'd like to note that I had a question was about the whether the spin of the doublet HSiO₂ was localized on the O₂ complex or not. I'm not certain where the author's were going with their response, but it would still be nice to know the location of the spin density on the i1 complex.

We modified our discussion and added the sentence in the first paragraph of the **Discussion** section:

The calculated spin density distribution in **[i1]** shows that the remaining unpaired electron in HSiOO is localized on the terminal O atom, which exhibits a spin density of 0.91 (Figure 3).

We also added the spin density distribution in Figure 3 to that the reader and the referee can visualize the spin density.

Reviewer #2 (Remarks to the Author):

My appreciation of the manuscript has not changed and I still recommend its publication, provided that the remarks below are taken into account. I especially advise the authors to be more rigorous in their choice of references.

Thank you.

- (lines 47-48) "Interstellar silicates also play a critical role in star formation and in the origin of solar systems contributing to the radiation balance and acting as a molecular feedstock⁷" the review article proposed as a reference concerns organic molecules in astrophysics. It does not provide the reader with information on the role of silicates in star formation and in the origin of solar systems, etc., even though it mentions silicate grains as components of cosmic dust and comets. Please refer the reader to relevant literature and clarify the meaning of "Interstellar silicates [act] as a molecular feedstock³". Are silicates decomposed into molecules? Or do they adsorb and then release molecules? What molecules are concerned? A feedstock for what process?

We modified this section to:

Interstellar silicates also play a critical role in star formation and in the origin of solar systems contributing to the radiation balance and acting as a molecular feedstock, both through the formation of complex organics through the release of icy mantles that cover them and through disruption of grains in interstellar shocks^{3,7}.

(lines 190-191) "At low pressures where collision deactivations are inefficient, the fragmentation channels take over": it would be good to give the current reference 61 as an example of third-body collisions, as in "Whereas third-body collisions prevent the dissociation of SiO₂ when the reaction Si + O₂ takes place in a 0.37-K superfluid He droplet⁶¹, the fragmentation channels take over at low pressures where collision deactivations are inefficient."

Changed.

- (line 193) "studies ... has firmly": studies ... have firmly

Changed.

- (lines 197-198) "... i.e. the reaction of silicon monoxide (SiO) with molecular oxygen (O₂), which plays an important role in fabrication of silicon oxide films at elevated temperatures^{46,47}": one would expect references 46 and 47 to describe an SiO₂-film production process that (i) involves the neutral-neutral SiO + O₂ reaction, (ii) requires a high temperature environment (a heated substrate, I guess), and (iii) is widely used, making the reaction important. I do not see how reference 47 is relevant, and I cannot verify reference 46 except for its abstract. In the latter

case, the following document is more accessible and likely equivalent: Ishikawa, Matsugatani, and Takaoka, *Vacuum* 39 (11–12), 1111-1113 (1989) [link: [https://doi.org/10.1016/0042-207X\(89\)91101-9](https://doi.org/10.1016/0042-207X(89)91101-9)]. The SiO₂-film production method it describes, which I expect to be the same as in reference 46, actually uses charged reactants, SiO⁺ and O₂⁺ (from an ICB source and a microwave ion source, respectively), not neutral species. It is also introduced as a method to produce good quality films while avoiding high temperatures. Therefore reference 46 is likely not relevant. Please refer the reader to literature on the role of the SiO + O₂ reaction in the fabrication of silicon oxide films at elevated temperatures.

We did literature search on the high temperature chemistry of SiO as suggested.

- (lines 236-237) "Hence, the recent detection of molecular oxygen toward Orion and ρ Oph A is truly extraordinary⁵³⁻⁵⁵": add Larsson et al. (2007) for the first certain observation of O₂ in ρ Oph A [link: <http://dx.doi.org/10.1051/0004-6361:20065500>].

We added this reference.

- (line 237) "Gas phase abundances of molecular oxygen higher than 10⁻⁶ would be ...": to be more precise, since reference 53 gives fractional abundances (relative to H₂) please write "Gas-phase abundances of molecular oxygen higher than 10⁻⁶ with respect to H₂ would be ..."

We changed it.

- (lines 237-239) "[abundances] higher than ... are significantly lower than ...": this is not logical. Here is an example with numbers: stating "abundances higher than 10 are in conflict with observations, but they are significantly lower than 100" does not make sense, because an abundance of 200 is higher than 10 and not lower than 100. Please rephrase the statement into "abundances in the range ... are higher than ..., but lower than ...", or "abundances of the order of ... are higher than ..., but lower than ...", for instance, with explicit values.

We rephrased the statement.

- (line 239) "... molecular oxygen abundance in interstellar ice⁵⁶": unless I am wrong, reference 56 does not give any definitive information on the abundance of molecular oxygen in interstellar ices and finally interprets observations assuming pure CO ices. At any rate, for a valid comparison, please refer to an article that gives the abundance of O₂ in interstellar ice with respect to that of H₂, as done for gas-phase O₂ in reference 53, or derive it explicitly from reference 56 so as to make your point clear to the reader. Maybe cite Vandebusshe et al. (1999) [link: <http://adsabs.harvard.edu/full/1999A%26A...346L..57V>] who give abundances of O₂ in the gas-phase and in ice relative to H (not H₂). Note that the abundance of O₂ in ice is not even an order of magnitude larger than its abundance in the gas phase according to these authors,

unless the entire oxygen deficit is assigned to O₂ ice; it is more likely contained in H₂O ice, however. Note that the recent search for O₂ by Wirström et al. (2016) [link: <https://doi.org/10.3847/0004-637X/830/2/102>] did not detect the molecule although the targeted regions were expected to be favorable. Upper limits for the fractional abundance of O₂ consistent with previous studies were determined ($N(\text{O}_2)/N(\text{H}_2) = (0.6-1.6) \times 10^{-7}$).

We changed it.

- (lines 250-254) "Both silicon dioxide and silicon monoxide – ... produce larger silicon oxides in exothermic chemistry³¹ – may eventually play a role in the formation and growth of interstellar silicates^{16,17}": according to their authors, references 16 and 17 concern the formation of silicates in circumstellar environments (of course it could be extended to the growth of silicates in the ISM, but the authors did not do it explicitly). These references must be inserted earlier in the sentence alongside reference 31: "Both silicon dioxide and silicon monoxide – ... produce larger silicon oxides in exothermic chemistry^{16,17,31} – may eventually play a role in the formation and growth of interstellar silicates." To propose references at the end of the sentence may not be necessary given that the formation (in the ISM) and especially the growth of interstellar silicates are phenomena often raised and speculated upon, but insufficiently studied experimentally to date.

We moved the references as suggested.

- (lines 256-258) "[silicon oxide plasmas] widely used in the semiconductor industry for depositing thin film insulators in integrated circuits like memory or processor chips⁶²⁻⁶⁴": references 63 and 64 deal with the chemical vapor deposition of diamond, which is not related to silicon oxide plasmas or the semiconductor industry (not yet at any rate). Please provide relevant references.

We changed this.

- (lines 280-283) "These pathways provide a population of silicon oxides, which can possibly provide a basis for the regeneration of interstellar silicates thus leading us closer to solving the paradox of the injection and destruction timescales of silicates through interstellar shocks^{71,72}": simply revert to the original sentence (without "through interstellar shocks") and use the references given in the introduction as in "... the paradox of the injection and destruction timescales of silicates^{5,21-24}."

We changed it.

In the experimental part, the reference for normalization of the counts (i.e., what the counts are normalized against) should be mentioned: - (lines 329-330) "we are able to record TOF spectra ... and normalize them, ..."

We modified the sentence as:

‘..., thus we are able to record TOF spectra at discrete angles, integrate and normalize them with respect to the intensity at the CM angle, ...’.

- (line 425, in the title of the vertical axis, in Figures 1A and 1B) "Normalized counts"

We changed it to: normalized experimental distribution.

- (line 491) in reference 18, add the missing initials: Plane, J. M. C. & Bromley, S. T.

We fixed it.

- (line 581) in reference 64, correct the name of the journal: "J. Phys.: Condens. Matter" (with colon and Matter, not "Mat.").

This reference now is not related in the current version of manuscript and deleted.

Reviewer #3 (Remarks to the Author):

I believe the manuscript can now be published in its present form.

Thank you. No revisions were requested.